# Harvesting Mango Fruit with a Short Stem-End Altered Endophytic Microbiome and Reduce Stem-End Rot

**DOI:** 10.3390/microorganisms8040558

**Published:** 2020-04-13

**Authors:** Ortal Galsurker, Sonia Diskin, Danielle Duanis-Assaf, Adi Doron-Faigenboim, Dalia Maurer, Oleg Feygenberg, Noam Alkan

**Affiliations:** 1Department of Postharvest Science of Fresh Produce, Agricultural Research Organization (ARO), Volcani Center, Rishon LeZion 7505101, Israel; ortalg@volcani.agri.gov.il (O.G.); soniad@volcani.agri.gov.il (S.D.); danielle.assaf@mail.huji.ac.il (D.D.-A.); daliam@volcani.agri.gov.il (D.M.); fgboleg@volcani.agri.gov.il (O.F.); 2Institute of Plant Sciences, Agricultural Research Organization (ARO), Volcani Center, Rishon LeZion 7505101, Israel; adif@volcani.agri.gov.il

**Keywords:** mango, stem-end rot, community, endophytic microbiome, antioxidant activity, antifungal activity

## Abstract

Stem-end rot (SER) is a serious postharvest disease of mango fruit grown in semi-dry area. Pathogenic and non-pathogenic microorganisms endophytically colonize fruit stem-end. As fruit ripens, some pathogenic fungi switch from endophytic colonization to necrotrophic stage and cause SER. Various pre/post-treatments may alter the stem-end community and modify SER incidence. This study investigates the effects of harvesting mango with or without short stem-end on fruit antifungal and antioxidant activities, the endophytic microbiome, and SER during fruit storage. Our results show that harvesting mango with short stem significantly reduced SER during storage. At harvest, fruit harvested with or without stem exhibit a similar microorganisms community profile. However, after storage and shelf life, the community of fruit without stem shifted toward more SER-causing-pathogens, such as *Lasiodiplodia, Dothiorella*, and *Alternaria*, and separated from the community of fruit with stem. This change correlated to the high antifungal activity of stem extract that strongly inhibited both germination and growth of *Lasiodiplodia theobromae* and *Alternaria alternata*. Additionally, fruit that was harvested with stem displayed more antioxidant activity and less ROS. Altogether, these findings indicate that harvesting mango with short stem leads to higher antifungal and antioxidant activity, retaining a healthier microbial community and leading to reduced postharvest SER.

## 1. Introduction

Mango (*Mangifera indica* L.) is an economically important fruit, which is generally cultivated in tropical, subtropical, and semi-dry regions and is distributed worldwide [1]. Because of the high humidity in tropical areas, the primary postharvest disease of mango fruit in those regions is anthracnose, which is caused by *Colletotrichum* sp. [2]. However, in Israel and other semi-dry areas, mango fruit is exposed to relatively low humidity and almost no rainfall from flowering to fruit development and harvesting. In recent years, probably one of the main postharvest diseases of mango fruit grown in semi-dry areas is stem-end rot, SER, due to global warming [3].

SER of mango is caused by a group of fungal pathogens, mainly members of Botryosphaeria genus, such as *Dothiorella dominicana*, *Dothiorella mangiferae*, *Lasiodiplodia theobromae*, *Neofusicoccum* spp., *Phomopsis mangiferae*, *Cytosphaera mangiferae*, and *Pestalotiopsis* sp., and by other pathogenic fungi, such as *Colletotrichum gloeosporioides* and *Alternaria alternata* [3,4,5]. The pathogenic fungi causing SER endophytically colonize mango pedicels and inflorescences [4,6]. After harvest, the fruit ripens, and many changes occur [7]. In response, the pathogenic fungi switch from the endophytic to a necrotrophic stage, which results in SER [4,6]. SER disease, which is caused by *Lasiodiplodia theobromae*, first appears as a small dark-brown area in the peel around the base of the fruit stem-end, which progresses into soft and watery decay radiating from the stem-end.

The mango stem-end is colonized not only by SER-causing pathogens, but also by various non-pathogenic species of microorganisms, including fungi, yeast, and bacteria [6]. The mango endophytic microbiome is versatile, and it is altered in response to exposure to sunlight and during storage at different temperatures [6]. Thus, we hypothesized that pre/post-harvest treatments would affect the endophytic microorganism community in the mango stem-end, which, consequently, will influence fruit-resistance and postharvest decay development, particularly SER.

Mango fruit has two natural abscission zones, one is located in the connection of the stem to the fruit (lower abscission zone), and the other is located about 0.5 cm above the fruit (upper abscission zone). Harvesting mango fruit with short stem-end (0.5 cm above the fruit) is a simple practice, which does not necessarily need the use of secateurs. When harvesting mango with the stem-end, more latex is retained in the fruit stem-end and the peel, as compared to the common practice of harvesting by detaching without stem [8]. Latex is distributed in mango fruit within a network of tiny ducts in the exocarp, the outer region of the mesocarp, and in the stem-end of the mango fruit [8]. This latex obtains a mixture of antifungal compounds, such as resorcinol derivatives, which are known to contribute to fruit resistance against fungal pathogens [9,10]. Thus, harvesting mango fruit with short stem-end should retain more latex with antifungal activity, which can reduce postharvest rot incidences.

Indeed, harvesting fruit with long stem was previously reported to reduce both anthracnose and SER development during ripening [11,12,13]. However, harvesting fruit with a long stem could be problematic for the packing-house application, as the stem can easily break and the latex can leak on the processing line. Therefore, harvesting mango fruit with short stem-end (approximately 0.5 cm, at the natural upper abscission zone) might be an alternative way to control SER disease. To the best of our knowledge, the effect of harvesting mango practice on the endophytic microorganism changes and their relation to SER occurrence during storage have never been studied before. In this study, we investigated the effect of harvesting mango with short stem-end when compared to harvesting without stem-end on the dynamics of fungal and bacterial microbiome endophytically colonizing mango stem-end and, as a result, on the incidences of SER during cold storage and shelf life.

## 2. Materials and Methods

### 2.1. Plant Material, Harvesting Practice, and Storage Conditions

Mango (*Mangifera Indica* L.) fruit cv. ‘Shelly’ was grown in the Galilee Sea region (northern Israel) under the same growth conditions, without any pre-harvest treatments, from the relatively dry spring (March) until the dry and hot summer (end of July). The fruits obtained 2 h after harvest from commercial orchard (‘Mor-Hasharon’ or ‘Eitan’ storage-house, Israel) and transported (less than 2 h) to the Agricultural Research Organization, Israel. Mango fruits were harvested while using two practices; with a short stem-end of 0.5 cm (‘with stem’- at the upper abscission zone) or by the common practice of detaching the fruit, which leaves no stem at all (‘no stem’—at the lower abscission zone). Uniform, unblemished fruit weighing ~340–400 g were selected, washed with tap water and air-dried, and were then placed into 4–5 cardboard boxes for each treatment. Separate cardboard boxes, each containing 10–12 fruits either with or without stem, were used for cold-storage experiments. Mango fruit was stored for three weeks at 12 °C in cold storage (CS) rooms and then transferred to 20 °C for seven days in order to mimic transportation and long shelf life (SL). The cold storage experiments were repeated in three consecutive seasons 2015, 2016, and 2017.

### 2.2. Evaluation of Fruit Quality Parameters in Mango Fruit

Physiological parameters—firmness index (1–10), total soluble sugars (TSS), and acidity (percentage citric acid equivalence)—of mango fruit were assessed at harvest, after cold storage, and after seven days of shelf life. Fruit firmness was determined by an electronic penetrometer force gauge (LT-Lutron FG-20KG, Indonesia) with an 11-mm probe, at two points on the central line of each fruit (10 measurements per treatment). The percentage of TSS was measured from the juice of fruit pulp using Palette digital-refractometer PR-1 (Model DBX-55, Atago, Japan), five fruits per treatment at each time point. The acidity was determined as a citric acid equivalent in 1 mL of pulp juice of mango that was dissolved in 40 mL of double-distilled water, while using an automatic titrimeter (Model 719 s, Titrino Metrohm Ion Analysis Ltd., Switzerland), five measurements per treatment in each time point.

### 2.3. Decay Evaluation of Mango Fruit

Percentages and severity (severity was assessed as index 0–10; zero- no decay, 1—mild decay, 5—moderate decay, 10—severe decay) of fruit with stem-end rot (caused mainly by *Lasiodiplodia theobromae*) or side rot (caused mostly by *Alternaria alternata*) evaluated for each box. Measurements were performed after three weeks in cold storage (12 °C) and after seven days of shelf life (SL) or ten days (prolong SL) of shelf life (20 °C). The decay evaluation experiments were repeated in three consecutive seasons 2015, 2016, and 2017, with similar results.

### 2.4. Fungal Material

Cultures of *Lasiodiplodia theobromae*, *Alternaria alternata,* and *Colletotrichum gloeosporioides* strains were grown on potato dextrose agar (PDA, Merck, Darmstadt, Germany) medium for 14 days at 28 °C. Conidia were gently collected, suspended in sterile distilled water, filtered through four layers of sterile cheesecloth, and diluted to a concentration of 10^6^ conidia mL^−1^. The conidial concentration in the suspension (number of conidia per ml) was microscopically determined while using a hemocytometer. The conidia suspension was used for the different antifungal activities, as described below.

### 2.5. Preparation of Mango Stem-End and Peel Extracts for Antifungal Activity

Total organic compounds in mango stem-end and mango peel were extracted, as follows. Stem-end and peel tissues were washed, dried, and submerged into liquid nitrogen, and were then ground to a fine powder using pestle and mortar. A 0.3 gr of powder was dissolved in 3 mL of extraction solution that contains methanol/water/HCl (80:20:0.1, *v*/*v*/*v*). Suspensions were extracted for 4 h on a shaker and centrifuged at 4000 rpm for 10 min. at 4 °C, and the resulted supernatants were stored at −20 °C. The pellet residue was re-suspended in a 3 mL extraction solution, extracted for 1 h, and then centrifuged, as described above. Subsequently, the organic fractions were combined and evaporated under vacuum to receive the organic extracts solutions. These extracts solutions served for the different antifungal activity tests and they were diluted, as described below.

### 2.6. Effect of Mango Stem-End Extract on Fungal Growth in Plates

The inhibition effect of mango stem-end extract against the hyphal growth of *Colletotrichum gloeosporioides* was tested on PDA plates. Liquid PDA (9.5 mL, 35 °C) was poured into Petri dishes that contained 0.5 mL of extract solution. The Petri dishes were shaken and allowed to solidify. Mycelium agar discs (4 mm diameter) were excised from the growing edge of 14 days culture of *Colletotrichum gloeosporioides* and were placed in the middle of the incorporated plates. The experiments were performed in triplicates. The plates were incubated for six days at 28 °C, and the hyphae diameter was measured.

### 2.7. Effect of Mango Stem-End and Peel Extracts on Fungal Growth in 96 Well-Plates

The antifungal activity of mango stem-end and peel extracts were tested against hyphal growth of *Lasiodiplodia theobromae*, *Alternaria alternata,* and *Colletotrichum gloeosporioides* by using an *in-vitro* kinetic absorbance assay. All of the assays were performed in 96-well plates. Each well contained 1000 conidia, 10 µL of stem-end or peel extracts (diluted 1:100), embedded in a total volume of 300 µL M3S liquid medium (Mathur’s medium containing the following reagents (per liter): 1 g of yeast extract (Difco laboratories, Inc., NJ, USA), 1 g of Bacto peptone (Difco Laboratories, Inc., Detroit, MI, USA), 10 g of sucrose, 2.7 g of KH_2_PO_4_, and 2.5 g of MgSO_4_·7H_2_O, and 250 mg of chloramphenicol, at pH 5.5). The positive control wells contained extraction solution (diluted 1:100). The plate was inserted into a Microplate Reader (EnSpire, 2300 multilabel reader, PerkinElmer, MA, USA) and incubated at 25 °C for absorbance measurements every hour, at 600 nm over 72 h. The absorbance of six wells of technical repeats was averaged together and was background-corrected by subtracting the average absorbance measured from wells containing only media at time zero. The results of the antifungal effect were represented as a fungal growth rate.

### 2.8. Effect of Mango Stem-End and Peel Extracts on Conidia Germination

The efficiency of stem-end and peel extracts were tested on *Lasiodiplodia theobromae* and *Alternaria alternata* conidia germination while using an *in-vitro* micro-assay method that was modified from [14]. Conidia suspensions were diluted to 10^5^ conidia mL^−1^ in sterile water. The extracts (10 μL) were added to a glass slide, containing 10 μL conidia suspension, and then incubated in a humid petri dish for 20 h at 25 °C. The conidia were examined under a microscope (Leica DM500 equipped with a Leica ICC50 HD camera, Wetzlar, Germany) to evaluate both germination’s percentage, and germ tube elongation. Conidia germination was determined when the germ tube’s length was equal to or more than the conidia diameter. The germ tube length was evaluated in three microscopic fields in each droplet, six droplets for each treatment, and determined using LasEZ Leica software (Wetzlar, Germany).

### 2.9. The Whole Fruit ROS Level

Harvested mango fruit with and without stem-end was used to measure the ROS level by fluorescence imaging, while using *in-vivo* Imaging System (IVIS Lumina II imaging system, PerkinElmer, USA). The ROS level was measured in mango stem-ends by staining with 10 µM of 2,7-dichlorodihydro fluorescein diacetate (H_2_DCF-DA, Sigma-Aldrich, St. Louis, MO, USA) in phosphate-buffered saline (PBSX1), and incubation for 15 min. in the dark, then the stem-ends were washed twice with PBSX1 before IVIS evaluation. The stained fruit was exposed to 488 nm excitation and 510 nm emission wavelengths. The relative intensity of the fluorescent signal was calculated using IVIS Lumina II imaging system software (PerkinElmer, USA), as the average green fluorescence in three biological repeats for each treatment with/without stem-end.

### 2.10. Antioxidant Activity

The radical-scavenging activity of mango stem-end extracts was performed in vitro by 2,2′-Diphenyl-1-picrylhydrazyl (DPPH, Sigma-Aldrich, St. Louis, MO, USA), according to the method described by [15], with slight modifications. For this, a solution of 0.1 mM DPPH was prepared. Stem-ends tissue samples (in fruit with a short stem) and bottom part of the stem-ends (in fruit without stem) were submerged into liquid nitrogen and they were ground to a fine powder using pestle and mortar. A 500 gr of powder was extracted with 3 mL of methanol/water solution (70:30 *v*/*v*). The suspensions were extracted for 4 h on a shaker and centrifuged at 4000 rpm for 10 min. at 4 °C, and the resulted supernatants were stored at −20 °C. The pellet residue was re-suspended in a 3 mL extraction solution and extracted for 1 h, and then centrifuged, as described above. The supernatants were combined and evaporated under a vacuum. These samples were diluted 1:100 with water for the assay. A 15 µL of diluted samples was added to 1 mL of DPPH solution and shaken well. The samples were kept in the dark for 30 min. A control was prepared with a 1 mL DPPH solution and 15 µL methanol/water solution (70:30 ***v***/***v***). The reaction was carried out in triplicate in 96-well plates using Microplate Reader (EnSpire, 2300 multilabel reader, PerkinElmer, MA, USA) for absorbance measurement at 517 nm. The scavenging activity was estimated based on the percentage of DPPH radical scavenging, as following:(1)$$\%\;\mathrm{radical}\;\mathrm{scavenging}\;\mathrm{activity}=\frac{\mathrm{Absorbance}\;\mathrm{of}\;\mathrm{control}-\mathrm{absorbance}\;\mathrm{of}\;\mathrm{the}\;\mathrm{sample}\;\times 100}{\mathrm{Absorbance}\;\mathrm{of}\;\mathrm{control}}$$

### 2.11. Microbiome Experimental Design

Healthy mango fruit was sterilized externally with 1% sodium hypochlorite solution for 2 min. and then twice washed with sterile water for 1 min. to remove epiphytic microorganisms. The stem-ends (0.5 cm long, in fruit with stems) and bottom part of the stem (0.2 cm, in fruit without stems) were sampled using a sterile surgical blade. A total of 60 stem-ends were collected over three sampling times (harvest, after cold storage, after shelf life). All of the samples were immediately frozen in liquid nitrogen and stored at −80 °C for further analysis.

#### 2.11.1. DNA Extraction

Stem-end tissue was freeze-dried (Christ alpha 1–2 freeze dryer, Osterode am Harz, Germany) for 5 h and ground using a mortar and pestle with liquid nitrogen. DNA was extracted following our previous CTAB protocol [6]. The protocol included incubation (60 °C, 10 min.) in 10% CTAB (*w*/*v*) and 5 M NaCl between chloroform: isoamyl purifications. The pellet was incubated with RNAse (0.05 mg per sample) at 37 °C for 30 min., and additional DNA precipitation in ethanol (3 M sodium acetate, pH 5.2) was performed. DNA quality and quantity were evaluated and measured on a gel and by a spectrophotometer (NanoDrop 1000 spectrophotometer, Thermo Scientific, Wilmington, DE, USA), respectively.

#### 2.11.2. Amplification and Sequencing

A targeted amplicon sequencing (TAS) approach was applied [16,17]. Briefly, two PCR stages were used to generate amplicons for Illumina MiSeq sequencing. First, 28 PCR cycles were performed with DNA and CS1 + CS2 primers containing 5′ linker sequences. The PCR amplification was validated by agarose gel electrophoresis. An aliquot of the PCR amplicon was transferred to a second PCR with new primer targets containing linker sequences for an additional eight cycles before sequencing on the Illumina MiSeq platform. All of the PCRs contained 20 ng DNA template, 1× KAPA HiFi Hot Start Ready Mix (KAPA Biosystems, Wilmington, MA, USA), and 0.6 pmol of each primer. Each PCR was performed three times, and the products were pooled. The amplicons were then submitted to DNA Services Facility (www.rrc.uic.edu/dnas) at the University of Illinois at Chicago (UIC) for a second PCR and Illumina MiSeq TAS according to the method described in [18].

For the bacterial microbiome, the V4 domain of bacterial 16S rRNA genes was amplified in a nested PCR to reduce plant organelle amplification. The first step was performed while using primers 515F (modified) (5′-GTGCCAGCMGCCGCGGTAA-3′) and 1062R (5′-CTCACRRCACGAGCTGAC-3′). The reaction conditions consisted of an initial 95 °C for 3 min. followed by 20 cycles of 98 °C for 20 s, 58 °C for 20 s, 72 °C for 1 min., and a final extension of 72 °C for 5 min. The second step was performed using 1:10 dilution of product from the previous step as the template, with the primers: CS1_515F (5′-ACACTGACGACATGGTTCTACAGTGCCAGCMGCCGCGGTAA-3′) and CS2_806R (5′-TACGGTAGCAGAGACTTGGTCTGGACTACHVGGGTWTCT-AAT-3′); the cycling conditions remained the same, with an annealing temperature of 52 °C and 5–8 cycle until a visible band could be detected on a gel with an expected length of 291 bp.

Fungal microbiome, internal transcribed spacer (ITS) were amplified with the primers CS1_BITS (5′-ACACTGACGACATGGTTCTACAACCTGCGGARGGATCA-3′) and CS2_B58S3 (5′-TACGGTAGCAGAGACTTGGTCTGAGATCCRTTGYTRAA-AGTT-3′). The reaction conditions consisted of an initial 95 °C for 3 min., followed by 28 (or less) cycles, until a visible band could be detected, of 98 °C for 20 s, 56 °C for 20 s, 72 °C for 1 min., and a final extension of 72 °C for 5 min.

#### 2.11.3. Data Analysis and Statistics

Raw Illumina FASTQ files were demultiplexed, quality-filtered, and analyzed using the QIIME toolkit [19]. Forward and reverse reads were merged using PEAR [20]. The merged FASTQ files were trimmed to generate high-quality data, as follows: (1) Ambiguous nucleotides (N) were trimmed from the ends, and reads with ambiguous internal nucleotides were discarded. (2) Primer sequences were trimmed from the reads, and any that lack either primer were discarded. (3) Reads were cut while using a quality threshold of *p* = 0.01. (4) Reads, after trimming, that were less than 300 bp and 100 bp in length were discarded for the 16S and ITS data, respectively. Chimeric sequences were identified using the UCHIME algorithm [21] when compared with the Silva 119 database [22] for the 16S data and compared with the UNITE ITS database [23] for the ITS data. Following quality filtering, the fungal ITS generated 1,846,153 reads (183 nucleotides mean length), and the bacterial 16S generated 4,573,888 reads (376 nucleotides mean length) (Appendix A).

Unique sequences were dereplicated from the combined sequences. Sequences with counts of greater than ten were used as seed sequences for sequence clusters. Low abundance sequences were queried against the master sequences while using USEARCH to find the master sequence with the highest percent identity with a cutoff of 98% and then added to the cluster. Operational taxonomic units (OTUs) were assigned using QIIME’s UCLUST [19] with a threshold of 98% sequence similarity. Reference-based OTU picking was performed using a representative subset of the Silva 119 reference database [22] and UNITE [23] as the fungal ITS database. Taxonomic annotations were assigned to each OTU using USEARCH and the reference database. The taxonomic and abundance data were generated for phyla, classes, orders, families, genera, and species.

Alpha-diversity (within-sample species richness) and beta-diversity (between-sample community dissimilarity) estimates were calculated while using QIIME weighted UniFrac distance [24] for bacterial 16S and fungal ITS reads. The principal coordinates were computed from the abundance matrices into the three-dimensional (3D) principal coordinate analysis (PCoA) plots using the R software. Co-occurrence correlation: Pearson correlations between fungi (family level) were calculated based on the abundance values. The correlation matrix was hierarchical clustered using heatmap3 R function. Heatmap of relative abundance of fungi: the dominant pathogenic fungi in each sample were selected from the results of species annotation and abundance information, and then clustered by their abundances (using heatmap3 R function). Co-occurrence correlations were performed: 1. Pearson correlations between fungi or bacteria were calculated based on the abundance values (family level). The correlation matrix was hierarchical clustered using heatmap3 R function. 2. Network graph inferred by using Conet plugin cytoscape software that was based on the Pearson correlation (family level) [25,26]. STAMP was performed to analyze the differences of the microbial community structure between the conditions while using Welch’s test *p*-value < 0.05 was considered to be statistical significance [27].

The data presented in bar graphs are an average and standard error. The T-test was performed to compare two treatments. One-way ANOVA analysis was performed by the Tukey–Kramer HSD test, and for non-parametric analysis by Kruskal–Wallis test using JMP Pro 14 software (SAS Institute, Cary, NC). Different letters or asterisks indicate significant differences (*p* ≤ 0.05).

## 3. Results

### 3.1. Harvesting Mango with Short Stem-End Reduce Postharvest Decays

To study SER in ‘Shelly’ mango, the fruit was harvested with a short stem of 0.5 cm length or following the common farmers’ practice by detaching the fruit and removing the stem at its base, which results in “no stem”. The fruit was stored for 21 days at 12 °C to mimic cold storage (CS), and an additional seven and 10 days at 20 °C to mimic shelf life (SL) and prolong SL, respectively. This prolonged shelf life enabled the development of stem-end rot (SER) and side rot in the fruit. Mango fruit harvested with or without stem showed relatively similar quality parameters during CS and SL, including decreasing firmness, increasing total soluble solids (TSS), and decreasing acidity levels (Table 1). While the stem-end length in mango fruit did not affect fruit ripening parameters (Table 1), fruit with stem developed substantially and significantly less SER and side rots than fruit that was harvested with no stems, after CS and more considerably after SL (Figure 1A–C). Moreover, in the fruit that was harvested with stem, SER onset was later than in fruit with no stems. These results were repeated on mango ‘Shelly’ following three consecutive seasons 2015–2017 (Table 1). Similar results were received in two other mango cultivars, “Noa” and “Kent” (Appendix A). In these mango cultivars, the fruit quality parameters of mango harvested with or without stem were also similar, during CS and SL, while fruit that was harvested with stem significantly reduced SER and side rots development, compared to fruit harvested without stem, after CS and SL (Appendix A).

### 3.2. Stem-End Endophytic Microbiome Modulations during Storage of Fruit Harvested with or without Stem

DNA was isolated from stem-end tissue samples (in fruit with a short stem) and bottom part of the stem-ends (in fruit without stem) of mango fruits at harvest, after 21 days at 12 °C CS, and after an additional seven days at 20 °C of SL, to study the microbiome alteration in the mango stem-ends during storage. Subsequently, the isolated DNA was amplified using fungal (ITS) and bacterial (16S) primers, deep sequenced, and analyzed bioinformatically and statistically. A total of 2,186,219 and 7,706,538 reads were generated for each ITS and 16S dataset, respectively (Appendix A). After the removal of low-quality reads and quality filtering, the fungal ITS generated 1,846,153 reads, and the bacterial 16S generated 4,573,888 reads, that were clustered into 1,916 and 1,009 total operational taxonomic units (OTUs; 98% nucleotide identity [ID]), respectively.

During prolonged storage, the fruits ripen, and a significant increase in the incidence of postharvest SER and side-decay (mainly caused by *A. alternata*) were observed in mango fruit that was harvested without stem compared to mango that was harvested with a stem (Figure 1). Concurrent to the increase in postharvest decay during storage, significant alterations during storage were detected in both bacterial and fungal communities of mangos harvested with and without stem-ends (Figure 2; Table 2; Appendix A). The bacterial and fungal community clusters of mango with and without stem-end during storage were compared while using principal component analysis (PCoA). At harvest, both of the endophytic bacterial and fungal communities, harvested with or without stems, were clustered together, while, after CS, the communities started to separate, and after SL, the communities were separated in a significant manner (Figure 2A,C; Table 2; Appendix A). PCoA analysis of the fungal community revealed that most of the samples in each time point were clustered, more or less, together in mango harvested either with or without stem-end. At harvest, most of the fungal samples of mango with and without stem-end were clustered together in the upper-left side of the graph. After CS, the samples of mango with and without stem-end could be seen on the right side of the graph, already distinct, but in close proximity to each other. Whereas, after 7 d of SL storage, the samples of mango with stem-end were clustered in the upper-right side of the graph, separately from the samples of mango without stem-end, which appeared in the lower-right of the graph (Figure 2A).

Similarly to the fungal community, in the PCoA plot of the bacterial community, the samples of mango harvested with or without stem at harvest were clustered together on the upper-middle side of the graph. After CS, the samples shifted to the upper-right side of the chart, mango with stem were separated, but clustered closely to the samples of mango without stem (Figure 2C). After SL, the samples of mango with or without stem significantly separated and appeared on the left side and the lower-right side of the graph, respectively (Figure 2C).

Taken together, these results indicate that the storage time is the major driving variable for the fungal and bacterial communities clustering, where, during storage, the harvesting practice of mango, with or without stem-end, clearly influenced the fungal and bacterial communities.

### 3.3. Microorganismal Community’s Changes in Mango that were Harvested with or without Stem-Ends during Storage

Many significant changes in the endophytic microbiomes of fungi and bacteria were identified during storage (Figure 2B,D; Appendix A). The total counts of fungal OTUs in both fruit with and without stems were the highest at harvest (67,592 and 49,792, respectively) and decreased following CS and SL (Figure 2B,D). However, when comparing fruit with stems to fruit without stems at CS (16,864 and 27,503, respectively) and SL (12,077 and 49,041, respectively), the total count of fungal OTUs in fruits without stems were higher than fruits with stems at each time point (Figure 2B,D), which correlated with increasing in SER incidences in fruit without stems after SL (Figure 1).

Further, we focused on the differences in the relative abundance of fungal taxa in mango fruit with and without stem-ends during storage. Fungal taxa that appeared to be the most abundant at harvest, Tremellaceae, almost disappeared after CS and SL. Other significant changes in the fungal microbiome across the time point included the increase in Dothioraceae (*Dothiorella*) and dramatically increase in Pleosporaceae (*Alternaria*) in the CS samples in both mango fruit with and without stems (Figure 2B; Appendix A). After SL, a significant difference was observed in the fungal endophytic microbiome abundance between the samples of mango fruit with stem to samples without a stem. The increase in the abundance of fungal pathogen families, Dothioraceae, and Pleosporaceae, mainly in samples without stem when compared to samples with stem, over time (Figure 2B) was correlated with the increase in SER incidences mainly occurring in samples without stems after SL (Figure 1B and Figure 2B).

Several changes were also observed in the relative abundance of the bacterial community (Figure 2D, Appendix A). The bacterial taxa that were the most abundant at harvest, mainly in mango fruit without stem, were Sphingomonadaceae. After CS, the abundance of this family reduced and replaced with Methylophilaceae, in both mango fruit with or without a stem. After SL, there was a significant difference between the samples with and without stem, and the Sphingomonadaceae family was significantly increased in samples without a stem (Figure 2D, Appendix A).

### 3.4. Pathogenic Fungi Dynamics in Mango Stem-End

We further investigated whether specific genus was responsible for the distinct difference since just a few families significantly contributed to the differences in the relative fungal abundance of mango fruit with and without stem after SL. The heatmap of relative abundance of pathogenic fungi in mango fruit harvested with and without stem over storage showed a major increase in these pathogens in samples harvested without stems, mainly after SL (Figure 3). Among the pathogenic fungi that significantly increased in those samples were *Dothiorella, Aureobasidium, Lasiodiplodia,* and *Aspergillus.* Other pathogenic fungi that were significantly enriched in the samples without a stem after CS were *Alternaria, Stemphylium,* and *Penicillium* (Figure 3). Most of these pathogens, such as *Dothiorella, Lasiodiplodia,* and *Alternaria*, are known to cause SER. Interestingly, some of those pathogens’ families co-occurred in the stem. For example, Trichomaceae (*Penicillium*), Dothioraceae (*Dothiorella*), and Montagnulaceae (*Montagnulaceae sp.*) endophytically co-occur in mango stems (Figure 4B, Appendix A). There were also bacterial groups with a positive correlation of co-occurrence, such as Entrobacteriaceae, Rhizobiaceae, and Sphingomonadaceae.

### 3.5. Mango without Stem increased Fruit ROS Level and Possess Low Antioxidant Activity

Harvesting mango fruit with or without stem-end causes a smaller or a larger wound area in the mango. This wound might affect the reactive oxidant species (ROS) production in the whole fruit. To inspect this hypothesis, fruits that were harvested with or without stem were stained with DCF to monitor fruit ROS levels. The accumulation of ROS was detected by an *in-vivo* imaging system (IVIS) after the DCF staining of mango stem 4 h after harvest. The ROS fluorescence intensity was significantly higher (1.5-fold) in mango that was harvested without stem in comparison to mango that was harvested with a stem (Figure 5A,B). This result suggests that harvesting mango with a short stem resulted in a smaller wounding and retained the stem that contains latex and other compounds with antioxidant properties, which reduced the ROS levels and enhanced fruit resistance. Antioxidant activity was assessed in the stem-end of mango fruit, in the mango that was harvested with and without stem after harvest, CS, and SL by DPPH. Indeed, a significant two-fold increase in the amount of scavenging activity of DPPH was seen in mango with stem as compared to mango without stem (Figure 5C). This suggested that fruit harvested with short stem possess a lower level of ROS, probably because of higher antioxidant level, as compared to fruit that was harvested without stem (Figure 5C). The differences between mango harvested with or without stem were retained during storage (Figure 5C).

### 3.6. Mango Stem-End Extract inhibits Pathogenic Fungal Growth

We extracted the total organic compounds from the stem (from fruit with stem) and the bottom part of the stem, ‘No stem’ (in fruit without stem) and tested *in-vitro* their antifungal activity against two main postharvest SER pathogens, *L. theobromae* and *A. alternata,* to comprehend the reduced occurrence of fungal pathogens in fruit harvested with stem.

Stem extracts display a strong inhibition of fungal growth rate and conidia germination of both of the SER pathogens, as compared to ‘No stem’ extracts (Figure 6). Stem extract inhibited *L. theobromae* and *A. alternata* conidia germination by 93% and 63%, respectively, after 20 h incubation on a slide (Figure 6B,C). The stem extract also inhibits the germ tube length of *L. theobromae* and *A. alternata* by 22% and 7%, respectively, after 20 h incubation on a slide (Figure 6B,C), while ‘No stem’ extracts did not inhibit the conidia germination (Figure 6B,C). In *in-vitro* fungal growth, we also found that *L. theobromae* and *A. alternata* were most inhibited in the presence of stem extract (from fruit after harvest), by 43% and 90%, respectively (Figure 6D). ‘No stem’ extracts (from fruit after harvest, CS and SL) display less inhibition against the fungal growth rates of both of the pathogens, as compared to the stem extracts (Figure 6D).

In addition, we found that the stem extract (embedded in PDA plates) significantly reduced the *C. gloeosporioides* fungal growth as compared to the control (Appendix A). The hyphal diameter of the pathogen was decreased by 50% on the plates that contain the stem extract after 132 h inoculation when compared to control (Appendix A). In another *in-vitro* assay, the stem extract also strongly inhibited *C. gloeosporioides* fungal growth rate (39%), when compared to the mild inhibition (20%) of the ‘No stem’ extract (Appendix A).

Surprisingly, we also found that the peel extracts, from fruit that were harvested with stem, show more inhibition against *L. theobromae* fungal growth rate and conidia germination, than peel extracts from fruit that were harvested with a stem (Appendix A). The percentage of germination of *L. theobromae* was also strongly inhibited by 100% in the presence of peel extract (from fruit with stem), and by 70% in the presence of peel extract (from fruit without stem), when compared to the control, after 20 h incubation (Appendix A). The peel extract (from fruit with stem) also inhibits the germ tube length of *L. theobromae* by 100% as compared to 80% inhibition by peel extract (from fruit without stem) after 20 h incubation (Appendix A). The fungal growth of *L. theobromae* was reduced by 24% in the presence of peel extract (from fruits with stem), and by 17% in the presence of peel extract (from fruit without stem), in comparison to the control (Appendix A). These results suggest that the antifungal components that exist in the stem are moving to the peel and increase the whole mango fruit resistance against fungal pathogens (*L. theobromae*, *A. alternata,* and *C. gloeosporioides)*.

## 4. Discussion

### 4.1. Effect of Harvesting Mango with Short Stem-End on SER Development during Storage

In Israel, the standard practice for harvesting mango fruit until 2016 included detaching the fruit without a stem. Interestingly, in recent years, there has been an increase in mango SER incidences in Israel and other relatively dry areas [3]. In the 2014 season, SER becomes a severe postharvest disease that caused a 30–40% loss of harvested mango fruit in Israel [7]. Therefore, the present study proposed an easily implemented practice of harvesting mango with a short stem to reduce the SER levels during storage. In this study, we showed that harvesting mango with short stem dramatically reduced SER development and side decay during storage, compared to fruit that was harvested without stem (Figure 1). These findings suggested that the stem plays a crucial role in SER resistance, and even a short stem that retains on harvested fruit can protect the fruit from postharvest decays. [12] also reported that mango fruit that was harvested with stem reduces anthracnose and SER development during ripening when compared to fruit that the stems were removed. [11] also received similar results and showed a reduction in anthracnose and SER incidence and severity in fruit with a long stem attached. In another study, the retaining of a stem on mango fruit was also shown to delay SER development [28]. Conversely, [4] reported a considerably higher severity of SER infection in fruit having stems attached than in those lacking stems and suggested that the retained stems might harbor endophytic SER pathogens longer than fruit without stems. Generally, all of these studies were focused on investigating the impact of harvesting mango with a stem on disease occurrences after storage. Our research is the first work determining the effect of harvesting practice (with or without stem) on the endophytic microbiota dynamics in mango fruit stems during storage.

### 4.2. Effect of Harvesting Mango with Short Stem-End on Microbiome Dynamics during Storage

We characterized the endophytic-microbiome of these samples during storage to understand the considerable differences in the postharvest SER occurrences between fruit harvested with and without stem. Many factors can be involved in determining the endophytic-microbiome composition of bacterial and fungal communities present in fruits. Fruit ripening is one of the most crucial factors that influence fruit microbiota diversity and dynamics as well as fruit susceptibility to postharvest decays. During ripening and storage, the mango becomes susceptible to fungal pathogens (Figure 1A, [27]). The fruit undergoes dramatic physiological changes during ripening, including increased bioavailability of sugars and amino acids and a decrease in the cell-wall strength and plant-defense mechanisms, which have a significant impact on fruit susceptibility to postharvest decay [7]. These physiological changes modify the endophytic microorganism’s environment, probably enabling the pathogenic fungi that present endophytically in the mango stem-end before fruit ripening [2,4], in order to switch to necrotrophic colonization during ripening and to cause SER. Indeed, we found significant changes in both fungal and bacterial communities during storage (Figure 2). Similarly, [6] also showed that during fruit ripening and storage, the endophytic microorganism’s environment in mango fruit significantly changes and, consequently, influences the fruit’s susceptibility to SER [6].

Among the changes that occur during storage and fruit ripening, we found that the Tremellaceae members decreased in both samples that were harvested with or without stem (Figure 2), some of its members are known as biocontrol yeast (for instance, *Cryptococcus laurentii*) and they were reported to reduce the postharvest disease of fruits [29]. On the contrary, members of the Dothioraceae and Pleosporaceae increased significantly from Harvest to CS in both mango fruit harvested with and without stems (Figure 2B); these families contain known SER pathogens of mango, such *Dothiorella* and *Alternaria*, respectively [2,4]. Pleosporaceae family (contains *Alternaria*), which is known to infect fruit in cold storage conditions, being presented in high proportion in the community after CS and significantly decreased after SL in both mango fruit with and without stems.

Besides the effect of fruit ripening on the endophytic microbiome, we showed that harvesting mango fruit with or without stem also has an essential impact on the stem-end microbiome, mainly after storage. Different management practices, such as hot water, biological control, and conditions as temperature or storage time can affect fruit microbiome community composition and dynamics [30]. In this study, we show that the fungal and bacterial communities’ profile of mango harvested with or without stems was very similar at harvest. While, after storage, there were differences in the fungal and bacterial communities between samples that were harvested with and without stems. The fruit that was harvested with stem maintaining a large percentage of non-pathogenic fungi. In contrast, the fruit that was harvested without stem hosted more pathogenic fungi (Figure 3).

Specifically, members of the Dothioraceae were more abundant in the community of fruit without stem when compared to fruit with a stem. This Dothioraceae family contains known mango SER pathogens as *Dothiorella dominicana* and *Dothiorella mangifera* [2,4]. Trichocomacea (contains *Aspergillus niger*) is another fungal family that was in a higher percentage in the community of fruit without stem compared to with stem, which might be used as an indicator of increased decaying processes. The endophytic bacterial community of mango with stem also differed from those of mango without stem, after storage. Rhizobiaceae that contains genus, such *Candidatus liberibacter*, significantly increased in the community of fruit without stem compared to fruit with stem in the bacterial family (Figure 2D), which is known as pathogenic gram-negative bacteria involved in postharvest decays [31]. All of these changes in the microbiome may explain the dramatic increase in SERs and side-rot incidences in mango harvested without stems compared to fruit with stem, after storage.

### 4.3. Antifungal and Antioxidant Properties of Mango Stem-End

The presence of the stem had a very significant effect on SER incidence and stem microbiome composition after storage. The differences in SER incidences and microbiota community between fruits that were harvested with and without stems could be derived from mango latex present in the stem-end, which might contribute to fruit resistance against SER. Mango latex appears to have direct involvement in the resistance of immature mango fruit to fungal infection [9,10]. The latex, when draining out, is known to separate into a non-aqueous (oily) phase and aqueous phase, in which both phases were reported to contain antimicrobial and antifungal activities. The aqueous phase is known to have significant chitinase activities that contribute to fruit resistance against SER [32,33], while the oily phase has previously been reported to contain, especially alk(en)ylresorcinols (5-n-heptadecenylresorcinol and 5-n-pentadecylresorcinol), which were proven to have antimicrobial and antifungal activities [34,35,36]. Here, we showed that mango stem extract exhibits antifungal activities that can inhibit the growth of mango fungal pathogens, such as *L. theobromae*, *A. alternata*, and *C. gloeosporioides* (Figure 6 and Appendix A). [11] proved that latex of unripe mango fruit contains a mixture of 5-(12′-heptadecenyl) and 5-pentadecylresorcinol, which are responsible for the resistance to *C. gloeosporioides* and *A. alternata*. [37] showed that mango peel extracts contain resorcinol and resorcinol derivative with antifungal properties. Differing relationships were found between the resorcinol levels and relative susceptibilities of different mango cultivars to anthracnose [11,37]. Strong positive relationships occur between resorcinol concentration in the peel and latex and fruit resistance to artificially inoculated anthracnose [12].

In this work, we also showed that stem extract from fruit harvested with stem contains high levels of antioxidants. Thus, as a result of the wounding during harvest, the fruit harvested with stem accumulated lower levels of ROS as compared to mango harvested without stem (Figure 5). Plant ROS plays an essential role in eliciting a wide range of defense mechanisms [38]. High ROS levels in plant cells result in a spreading cell death, which provides nutrients to necrotrophic pathogens. While a low concentration of ROS can act as a signaling molecule, usually leading to host protection against necrotrophs [39]. This response can contribute to the activation of fruits effective resistance against postharvest fungal pathogens, which are all necrotrophic during their aggressive stage [7]. These results provide another evidence that harvesting mango with stem, which possesses antioxidant activity and antifungal activity, is important in enhancing fruit resistance against postharvest decay.

## 5. Conclusions

The results of this study demonstrate that the agro-technical procedure of harvesting mango with a short stem that contains antifungal activities and contains antioxidants has a major effect on the fruit endophytic microbiome dynamics during storage. Thus, harvesting with a short stem supports the presence of non-pathogenic communities over pathogenic fungi in the fruit endophytic microbiome, which, consequently, leads to a reduction of SER in stored mango fruit.

## Figures and Tables

**Figure 1 microorganisms-08-00558-f001:**
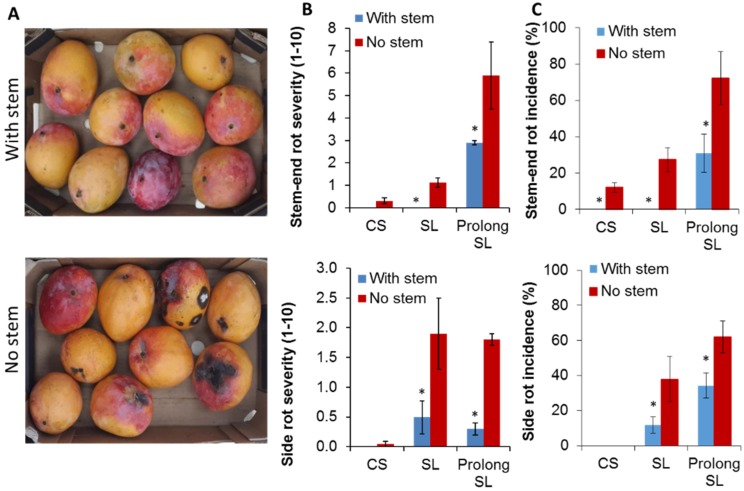
Effect of harvesting practice on stem-end rot and side rot incidences, and endophytic fungal community distribution. ‘Shelly’ mango fruit was harvested with (0.5 cm) or without stem (No stem) and cold-stored for 21 days at 12°C (CS) followed by seven days shelf life (SL) or ten days (prolong SL) at 20 °C. (**A**) Representative pictures of ‘Shelly’ fruit harvested with (upper section) or without stem (lower section) after shelf life. (**B**) Stem-end rot severity (upper section) and side rot severity (lower section), 2016 season. Rot severity was assessed by index 0–10: zero—no decay, 1—mild decay, 5—moderate decay, 10—severe decay. (**C**) Percentage of stem-end rot incidence (upper section) and side rot incidence (lower section), 2016 season. Average and SE are presented. Asterisk (*) indicates a significant difference between samples with and without stem in each of the time points (*p* ≤ 0.05).

**Figure 2 microorganisms-08-00558-f002:**
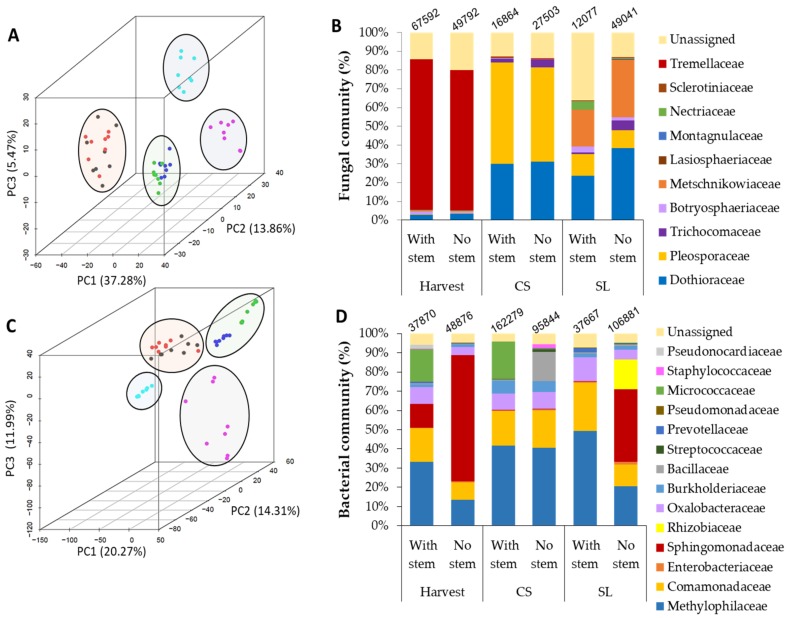
Dynamics of endophytic microbial communities in mango fruit harvested with or without stem-ends during cold storage (CS) and shelf life (SL). (**A**) PCoA hierarchical cluster UniFrac distance of the fungal community. (**B**) Stacked column charts representing a taxonomic community of fungi (family level). (**C**) PCoA hierarchical cluster UniFrac distance of bacterial communities. (**D**) Stacked column charts representing a taxonomic community of bacteria (family level). In (**B**) and (**D**) above the stacked column, the operational taxonomic units (OTUs) numbers are represented.

**Figure 3 microorganisms-08-00558-f003:**
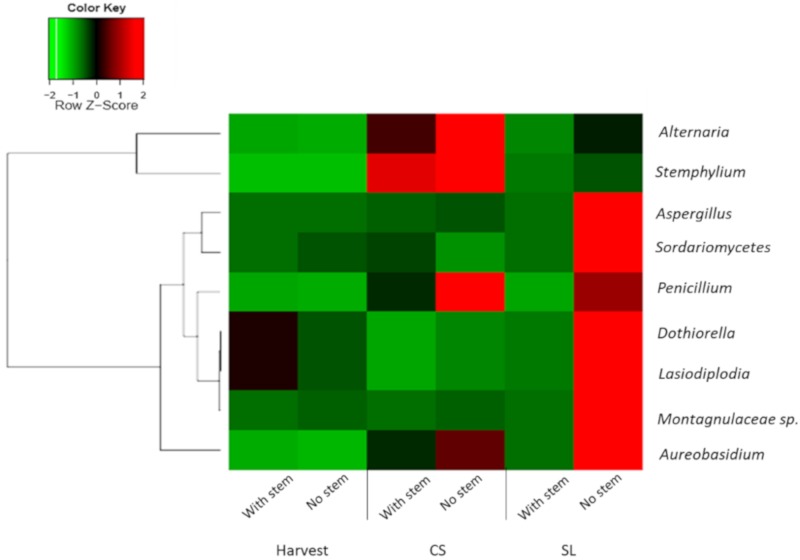
Heatmap of relative abundance of pathogenic fungi (genera level) that live endophytically in the mango stem in fruit that was harvested with or without stem-ends during harvest, cold storage (CS), and shelf life (SL). ‘Shelly’ mango fruit was harvested with or without stem and cold-stored for 21 days at 12 °C (CS), followed by seven days shelf life (SL) at 20 °C. The nine fungal pathogens significantly enriched in mango harvested without a stem after SL (*p* < 0.001).

**Figure 4 microorganisms-08-00558-f004:**
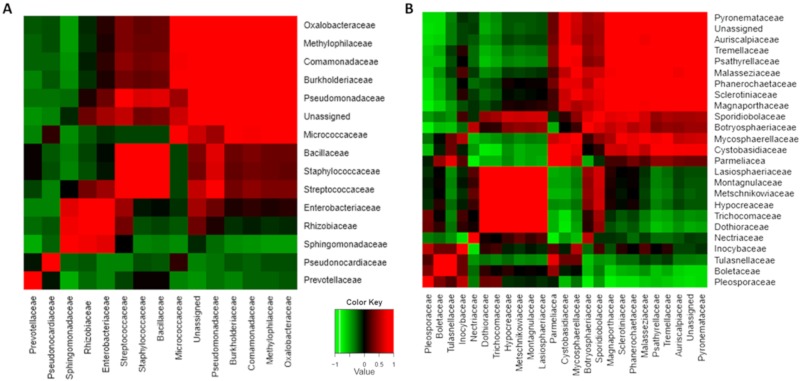
Co-occurrence heatmap patterns of fungal and bacterial microbiota families that live endophytically in the mango stem-end (family level). (**A**) Co-occurrence heatmap for bacterial microbiome. (**B**) Co-occurrence heatmap for fungal microbiome. The green and red colors indicate significant positive and negative correlations between two families, respectively.

**Figure 5 microorganisms-08-00558-f005:**
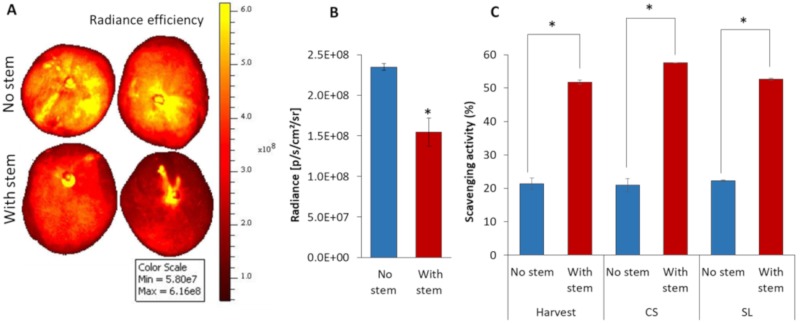
Effect of harvesting practice on fruit ROS level and antioxidant activity in fruit harvested with or without a stem. Fruit that was harvested with or without stem was stained with DCF to monitor fruit ROS. (**A**) Representative fluorescent picture of fruit harvested without stem (upper section) or with a stem (lower section). (**B**) Reactive oxygen species (ROS) levels in mango harvested with and without stem (as expressed by radiance efficiency). (**C**) Antioxidant activity in the stem (in fruit with stem) and bottom part of the stem (in fruit without stem) during cold storage (CS) and shelf life (SL), expressed by percentage of scavenging activity.

**Figure 6 microorganisms-08-00558-f006:**
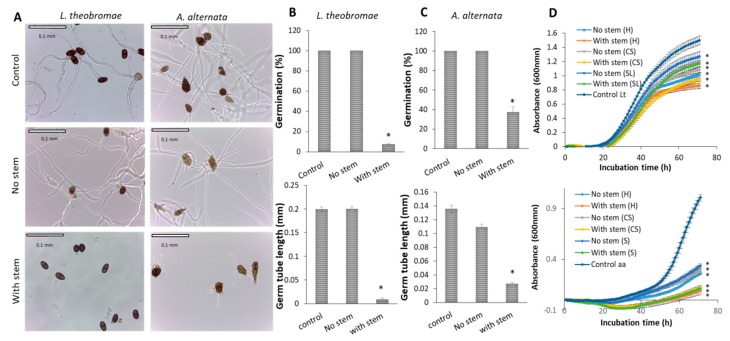
Effect of mango stem-end extracts on *L. theobromae* and *A. alternata* germination and growth. Stem-end of ‘Shelly’ mango fruit was organically extracted for evaluating the antifungal activity against fungal pathogens *L. theobromae* and *A. alternata*. The effect of different extracts (in 1% methanol) on conidia germination was evaluated by microscopic determination after 20 h of incubation. (**A**) Microscopic images (40× magnification) showing *L. theobromae* and *A. alternata* germination inhibition by stem extracts compared to control. (**B**) Percentage germination (upper section) and germ tube length (lower section) of *L. theobromae. (***C**) Percentage germination (upper section) and germ tube length (lower section) of *A. alternata*. (**D**) In vitro growth rate of *L. theobromae* (upper section) and *A. alternata* (lower section) in microplates in the presence of the different extracts. Each sample was averaged and background-corrected. Data presented as mean ± SE (*n* = 6). Asterisk (*) represents a statistically significant difference between each extract and control, using t-test, *p* ≤ 0.001. H; harvest, CS; cold storage and SL; shelf life.

**Table 1 microorganisms-08-00558-t001:** Evaluation of fruit quality parameters and SER incidence in “Shelly” mango fruit harvested with or without stem, after three weeks in cold storage at 12 °C (CS) and followed by seven days of shelf life storage (SL), or ten days of prolonging shelf life storage (prolong SL) at 20°C, following three consecutive seasons (2015–2017).

		2015		2016		2017	
	Storage Time	No Stem	With Stem	No Stem	With Stem	No Stem	With Stem
TSS (%)	CS	11.7 ± 0.30	13.7 ± 0.55	12.7 + 0.4	12.8 + 0.5	12 + 0.75	11.8 + 0.46
	SL	14.5 ± 0.35	14.2 ± 0.38	12.1 + 0.6	11.8 + 0.3	12.9 + 0.22	12.2 + 0.18
Acidity (%)	CS	0.90 ± 0.04	0.95 + 0.02	0.8 + 0.06	1 + 0.03	0.6 + 0.01	0.5 + 0.02
	SL	0.27 ± 0.06	0.33 ± 0.01	0.47 + 0.03	0.5 + 0.03	0.3 + 0.05	0.2 + 0.04
Color change (index 1–10)	CS	3.2 ± 0.1	2.8 ± 0.1	2.6 + 0.2	2.3 + 0.1	4 + 0.19	3.3 + 0.27
	SL	7.9 ± 0.5	7.7 ± 0.6	8.7 + 0.4	9.9 + 0.1	9.5 + 0.11	9.2 + 0.13
Firmness (index 1–10)	CS	10 ± 0.1	10 ± 0.0	9.9 + 0.1	10 + 0.1	9.6 + 0.2	10 + 0.0
	SL	5.4 ± 0.2	5.2 ± 0.2	3.8 + 0.2	4 + 0.0	3.5 + 0.5	3.7 + 0.3
SER (%)	CS	0	0	12.0 + 2.7	0	0	0
	SL	0	0	27.3 + 6.7	0 *	8.1 + 2.6	0
	Prolong SL	82.3 + 8.3	29.9 + 7 *	72.2 + 14.7	31 + 10.5 *	23.8 + 1.3	0 *

Fruit quality parameters: Fruit Brix (%TSS, total soluble solids), Acid (%citric acid equivalence), color change index (1–10), firmness index (1–10), and SER incidence (%). Asterisk (*) represents a statistically significant difference between samples with and without stem in each year, (*p* ≤ 0.05).

**Table 2 microorganisms-08-00558-t002:** Statistical analysis of category effect on microbial Alfa-diversity conducted by Shannon Index.

**Fungal ITS**			
**Treatment**	**Comparison**	**H**	***p*-value**
With stem	Harvest/CS/SL	11.29	0.0007
Without stem	Harvest/CS/SL	10.6	0.0011
Harvest	With stem/without stem	0.71	0.401
CS	With stem/without stem	4.41	0.036
SL	With stem/without stem	9.93	0.001
**Bacterial 16S**			
**Treatment**	**Comparison**	**H**	***p*-value**
With stem	Harvest/CS/SL	9.28	0.002
No stem	Harvest/CS/SL	0.54	0.462
Harvest	With stem/No stem	2.82	0.09
CS	With stem/No stem	11.29	0.0007
SL	With stem/No stem	0	1

CS; Three weeks in cold storage at 12 °C. SL; Seven days shelf life storage at 20 °C. Harvesting with a stem (0.5 cm) or without stem (No stem), (*p* ≤ 0.05).

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
