# Peer review of "Harvesting Mango Fruit with a Short Stem-End Altered Endophytic Microbiome and Reduce Stem-End Rot"

_microorganisms, 2020, doi:10.3390/microorganisms8040558_

Round 1

Reviewer 1 Report

The manuscript title “Harvesting mango fruit with a short stem-end altered endophytic microbiome and reduce stem-end rot” covered the significant information about the endophytic microbiome structure and dynamic against the stem end rot causing by Lasiodiplodia, Dothiorella, and Alternaria. This change correlated to the high antifungal activity of stem extract that strongly inhibited both germination and growth of Lasiodiplodia theobromae and Alternaria alternata. Higher antioxidant activity and less ROS in harvested fruit correlated with the data and hypothesis. However, the study sound a primary study and several technical and experimental issues need to fix.

  1. In result section sequencing data missing such as how many total sequence obtained, how many qualified tag sequences determined, total number of OTUs distribution.
  2. Table 2 the p values are normal or adjusted p value?
  3. Please add PERNOVA p value in figure 2 A (PcoA) and C (PcoA).
  4. Figure 2 B and D: which taxa have shown significant change, please add in supplementary.
  5. Figure 4 Co-occurrence heatmap patterns shows less informative, please add a network graph by using either Conet plugin cytoscape software (https://www.ncbi.nlm.nih.gov/pmc/articles/PMC5089131/) or Gephy https://www.ncbi.nlm.nih.gov/pmc/articles/PMC6046413/ that can indicates the appropriate interaction of microbiome.

Author Response

Answers:

  1. We added in lines 331-335 in the result section description about data sequencing and filtering, we also added to the supplemental two tables table S2 and S3 of sequence statistics of ITS and 16S libraries, respectively, after read processing.
  2. The values in table 2 are normal p value.
  3. We added the percentage of variation explained in the microbial community in each axis of figure 2 A (PcoA) and C (PcoA).
  4. We added to the supplementary table S4 with significant fungal and bacterial families across the condition in the current study. In accordance the table was added to the text in the correct place (line 383, 384, 395, 400 etc.')
  5. We think that the heatmaps is not less informative than the network graphs and it is easier to read. However, based on your request, we added to figure 4 (the co-occurrence heatmap) a networks graphs at Figure S2.

Reviewer 2 Report

These are my main comments on the MS (sensors-703087) entitled:“ Harvesting mango fruit with a short stem-end altered endophytic microbiome and reduce stem-end rot”

It is a very interesting and useful study investigating the presence or absence of short stem of harvested mangos on the presence of various pathogens and especially of SER disease. The study is generally very well organized and presented. My suggestion is to be published after minor revision. Authors may follow my suggestions (especially on M&M section) in order to improve their MS before publication.

Comments

Par 2.1. Where were these mangos cultivated (region, coordinates etc)? I think that it is important to prove that experimental fruits were cultivated under similar conditions (pre-harvest treatments, cultural practices, location of trees, crop variety etc). Many pre-harvest factors many influence the presence of post-harvest diseases in fresh fruits.

Par 2.1.Were there any specific criteria to characterize the decay as moderate or mild? The severity index is not very clear. Authors should clarify the way they characterized a decay as 5 or 6 or 7, for example.

Par. 2.4. Please describe the procedure to estimate conidial concentration (haemocytometer?).

Figures. The Figs are very well constructed. My only slight objection is that some of them contain many data and it is difficult to follow. Maybe separating some of them would improve the presentation (eg. the microbiome community from fig1).

Author Response

Answers:

Par 2.1. This comment is very important, we added to the manuscript, in line 78-80, information about mango growth, and emphasized that the fruit were cultivated under same conditions.

Par 2.1 We characterized the side and stem-end decays in the manuscript using 2 common ways: percentage and severity rot. The severity rot was assessed as severity index, which range between 0-10: zero- means no decay, 1-mild decay, 5-moderate decay, 10-severe decay. We added this in Par 2.3 and lines 316-318.

Par. 2.4. The conidial concentration (number of conidia per ml) was microscopically determined using a haemocytometer. We added to the manuscript, in line 113-114 the description of procedure of conidial concentration measurement.

Figures. We agree that separating the figures might simplify the presentation. However, we think that it is not recommended to separate the figures, because we already have six figs. that all include essential data for the manuscript.

Reviewer 3 Report

The reviewed manuscript is focused on the beneficial effect of simple agro-technical procedure of harvesting mango with a short stem for maintaining fruit quality during storage. The research presents the complex mechanisms of mango’s stem chemical compounds on endophytic microbiome dynamics during storage. The Authors used different and complex analytical techniques to verify research hypotheses. The manuscript clearly written and well organised, findings were discussed correctly with the literature cited. The manuscript presents novelty and suits well with ecological management in horticulture.

I propose to accept the manuscript with minor revision, concerning mainly spelling mistakes.

Author Response

The manuscript was fixed and the English was corrected throughout the manuscript.